# Influence of Topographic Characteristics on the Adaptive Time Interval for Diffusion Wave Simulation

**Pin-Chun Huang [1], Kwan Tun Lee [1,2,\*] and Boris I. Gartsman [3]**

[1]  Center of Excellence for Ocean Engineering, National Taiwan Ocean University, Keelung 20224, Taiwan; pinchunhuang@gmail.com

[2]  Department of River & Harbor Engineering, National Taiwan Ocean University, Keelung 20224, Taiwan

[3]  Institute of Water Problems of Russian Academy of Sciences, Moscow 119333, Russia; gartsman@inbox.ru

**\***  Correspondence: ktlee@ntou.edu.tw; Tel.: +886-2462-2192-6121

**Abstract:** Frequent flash floods in recent years have resulted in a major impact on the living environment, urban planning, economic system and flood control facilities of residents around the world; therefore, the establishment of disaster management and flood warning systems is an urgent task, required for government units to propose flood mitigation measures. To conserve the numerical accuracy and maintain stability for explicit scheme, the Courant–Friedrich–Lewy (CFL) condition is necessarily enforced, and it is conducted to regulate the relation between the numerical marching speed and wave celerity. On the other hand, to avoid the problem of flow reflux between adjacent grids in executing 2D floodplain simulation, another restriction on time intervals, known as the Hunter condition, was devised in an earlier study. The objective of this study was to analyze the spatial and temporal distribution of these two time-interval restrictions during runoff simulations. Via a case study of the Komarovsky River Basin in Russia, the results show that at the beginning of a storm, the computational time interval is restricted by the CFL condition along the upstream steep hillsides, and the time interval is subject to the Hunter condition in the mainstream during the occurrence of the main storm. The reason of a reduction in computational efficiency, which is a common problem in conducting distributed routing, was clearly explained. To relax the time-interval restrictions for efficient flood forecasting, the research findings also indicate the importance of integrating modified hydrological models proposed in recent studies.

**Keywords:** flood simulation; diffusion wave; time interval; watershed topography; runoff forecasting system

## 1. Introduction

Typhoons or torrential rains have become more frequent in recent years, hence, more careful and thorough consideration for establishing flood warning system is needed to alleviate flood disasters. Owing to the complete development of digital topography treatment [1–3], the grid-based digital elevation model (DEM) datasets have been applied to a variety of study fields. Although at present high-resolution elevation datasets are readily available and various software tools can be used to automatically extract geomorphological factors of watersheds, the computational performance of executing distributed runoff simulations remains to be improved to effectively exploit a finer dataset with a huge number of grids. The merit of the distributed runoff model is its capability to fully grasp the spatial and temporal variation in runoff transport by solving the hydrodynamic equations using numerical schemes. The simplified momentum equation based on a non-inertia wave (diffusion wave) type, which neglects the local and convective acceleration terms in the momentum equation,

has been widely applied because of its simple mathematical operations [4–8]. Several studies have shown that the simplified non-inertia wave equation can still provide appropriate solutions within tolerable errors in overland-flow simulations even for a rugged landform [6,9]. On the other hand, in comparison to the fully 2D model, quasi-2D models may neglect some significant aspects of the spatial variability of hydraulics and are too simplistic in the treatment of flow paths; however, owing to recent developments in topographic measurement for high resolution data, many previous studies have noted a growing prospect for quasi-2D models applied to the floodplain simulation [10–12]. In addition, quasi-2D model is also able to perform flood inundation simulation with unstructured grids [13]. Nevertheless, a restriction in time step needs to be regulated to avoid the crash or oscillation in numerical schemes.

Computational efficiency and numerical stability are usually acknowledged as the two primary concerns in conducting distributed runoff routing, though these are difficult to balance because the computational speed is subject to the time step restrictions, which are obeyed to validate the numerical schemes. Basically, there are two types of restriction on time interval for grid-based routing system. The well-known Courant–Friedrich–Lewy (CFL) condition [14] is adopted to adjust the numerical marching speed ($\Delta x / \Delta t$), which has to be higher than the wave celerity to ensure the numerical stability for explicit scheme. The other is the Hunter condition [8,12], which was proposed to avoid back-and-forth refluxing between adjacent grids, resulting in a so-called "checkerboard oscillation" (Hunter et al., 2005; Bates et al., 2010). Flow refluxing usually occurs on floodplain areas with a fairly gentle ground bed.

To reinforce the computational performance, several modified approaches, which intend to relax either of the two restrictions on time intervals, have been proposed [8,12,15–17]. Nevertheless, it is still worthwhile to elaborate upon how these two restrictions change with the topographic characteristics and runoff states in a watershed, so as to realize the necessity of applying an integrated model for runoff simulations. This study aimed to analyze the spatial and temporal distribution of the two time-interval restrictions in executing runoff routing. To avoid the influence of data resolution on the limited time step, related simulation cases were performed on a study watershed using structured data with fixed resolution. The magnitude of the allowable time interval and the difference between these two restrictions under various topographic and hydrological conditions are illustrated in detail. The research findings are expected to show the dominant cause of the decrease in computational efficiency when conducting distributed runoff routings, and further to demonstrate the significance of adopting an integrated algorithm recently published [18]. It should be noted that this study only focus on the time-step analyses for diffusion wave model based on structured grids, hence, the applicability of modified algorithm adopted in an unstructured data still needs to be discussed in the future research.

## 2. Approach for Distributed Runoff Routing

The de Saint Venant equations, according to the physically based hydrodynamic theorem, are usually applied to distributed overland-flow routings. In this study, the determination of flow direction for each grid follows the steepest gradient of the water surface among eight adjacent grids; in other words, the flow discharge of each grid can merely be assigned to its single downstream grid. Therefore, only a one-dimensional (1D) equation is required to calculate the flow discharge in a single direction and to conduct a two-dimensional (2D) grid-based overland flow simulation. Such a method to perform runoff simulation is termed quasi-2D routing. This system concept has been implemented for distributed watershed routing in several previous studies [6,13]. If the wind shear, eddy losses, and the *x* component of the rainfall intensity are neglected, the governing equations in a 1D form consisting of a set of continuity and momentum formulations can be expressed as follows:

$$\frac{\partial h}{\partial t} + \frac{\partial q}{\partial x} = i_e \tag{1}$$

$$\frac{\partial q}{\partial t} + \frac{\partial}{\partial x}\left(\frac{q^2}{h}\right) - gh\left(S_o - \frac{\partial h}{\partial x}\right) + ghS_f = 0 \tag{2}$$

where $h$ is the water depth, $q$ is the discharge per unit width, $i_e$ is the intensity of excess rainfall, $S_o$ is the ground slope, and $S_f$ is the friction slope. By neglecting all the inertia terms in the momentum equation, a simplified momentum equation in a non-inertia wave type, adopted to derive the discharge of each cell, can be expressed as follows:

$$q = \frac{1}{n}h^{5/3}S_f^{1/2} \tag{3}$$

where

$$S_f = S_o - \frac{\partial h}{\partial x} \tag{4}$$

where $n$ is the Manning roughness coefficient. Equation (3) indicates that the movement of flow is dominated by the gradient of the water surface. The conventional explicit scheme used to solve the aforementioned governing equations is explained in the following section.

### 2.1. Conventional Numerical Scheme

At the beginning of the simulation, the initial flow depth and grid elevation are used to calculate the water surface elevation; then, the single flow direction of each grid can be determined according to the steepest water surface gradient among the adjacent grids. The gradient of the water surface can be provided for Manning's equation to obtain the flow discharge. Subsequently, following the first-order backward finite difference scheme, the continuity equation as shown in Equation (1) can be discretized to estimate the time-varying increment of the water depth as follows:

$$\Delta h_j = \frac{\Delta t}{\Delta x}\left[q_{j,IN}(t) - q_j(t)\right] + \frac{\Delta t}{2}\left[i_e(t) + i_e(t + \Delta t)\right] \tag{5}$$

where $q_j(t)$ denotes the discharge per unit width of grid $j$ and $q_{j,IN}(t)$ represents the total inflow discharge per unit width of grid $j$, namely the accumulated discharge collected from all the upstream adjacent grids. For example, as shown in Figure 1, the inflow discharge of Grid 5 can be expressed as follows:

$$q_{5,IN}(t) = q_1(t) + q_2(t) + q_4(t) \tag{6}$$

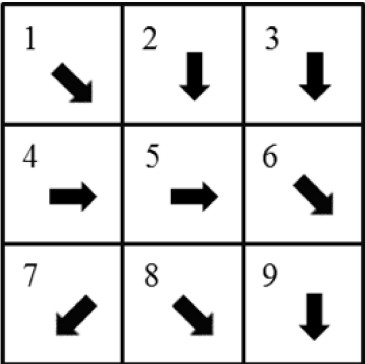

**Figure 1.** Drainage directions of grid-based dataset in quasi-2D routing system.

Hence, the water depth at $t + \Delta t$ can be obtained as follows:

$$h_j(t + \Delta t) = h_j(t) + \Delta h_j \tag{7}$$

By substituting the updated water depth into Equations (3) and (4), the discharge at $t + \Delta t$ can be expressed as follows:

$$q_j(t + \Delta t) = \frac{h_j(t + \Delta t)^{5/3}}{n} \left( S_o - \frac{h_j(t) - h_{j,DOWN}(t)}{\Delta x} \right)^{1/2} \tag{8}$$

where $h_{j,DOWN}(t)$ represents the downstream water depth of grid $j$. For example, as shown in Figure 1, Grid 6 is recognized as the single downstream grid of Grid 5. The aforementioned content illustrates the conventional procedure of DEM-based non-inertia wave simulation using an explicit scheme. It should be noted that to consider a possible backwater effect the flow direction of each grid has to be re-determined at each time step following the time-varying water surface level.

### 2.2. Time Interval Restrictions for the Conventional Scheme

To conserve the accuracy of numerical solutions and maintain the stability of the numerical schemes, two types of conditions are necessary to provide an allowable time interval during the process of a runoff simulation. One is the CFL condition [14] and the other is the Hunter flow limit condition [12]. In executing explicit schemes, the CFL criterion is acknowledged as the stability condition of the non-inertial wave model [6,19], whose time-interval restriction, depending on the grid size and wave celerity, can be expressed as follows:

$$\Delta t_{MC} = \frac{\Delta x}{c} \tag{9}$$

where $\Delta t_{MC}$ is the time step subject to the CFL criterion and $c$ is the celerity of non-inertia wave, which can be expressed as $(dQ/dh)/\Delta x$ [20,21]. The CFL condition is believed to validate the numerical algorithm; nevertheless, it is still not sufficient to completely guarantee the stability [8,12].

When the 2D non-inertia wave simulation is performed on a nonstaggered grid-based routing system, the problem of a "checkerboard oscillation" usually occurs [8,12,22]. This oscillation is induced by excessive flow volume leaving from a cell in a single time interval and causing flow reflux during the next time step as illustrated in Figure 2. This situation is particularly prone to occur on floodplains in rainfall-runoff simulations. To avoid overestimation of the solution and to provide an unreasonable constraint on discharge delivering to the downstream grid, Hunter et al. [12] deduced a criterion for determining an adaptive time step as follows:

$$\Delta t_{MH} = \frac{\Delta x^2}{4} \left[ \frac{2n}{h^{5/3}} \left| \frac{\partial h}{\partial x} \right|^{1/2} \right] \tag{10}$$

where $\Delta t_{MH}$ is the allowable time step subjected to the Hunter condition. It can be inferred that the adaptive time step shown in Equation (10) would decrease with a decreasing grid size $\Delta x$ and water surface gradient. The two aforementioned criteria have to be used at each time step during the flow simulation.

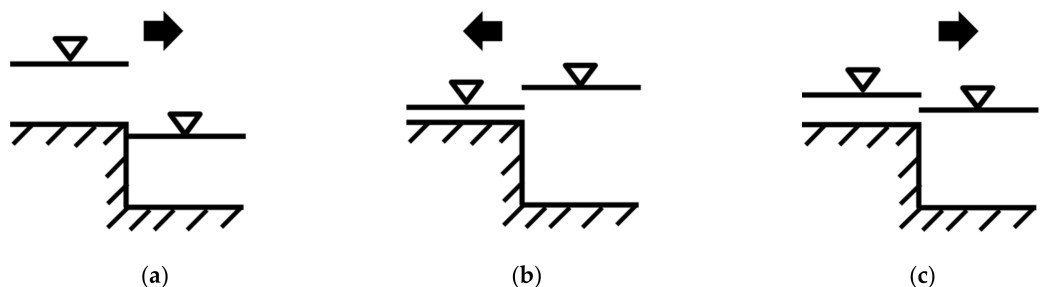

**Figure 2.** Schematic diagrams denoting the flow reflux between adjacent grids; (**a**) 1st time step; (**b**) 2st time step; (**c**) 3st time step.

*2.3. Rainfall Infiltration Estimation*

In this study, runoff simulation of storm events caused by excessive rainfall is the main concern. The Green-Ampt method [23] was adopted to deduct the amount of infiltration from the rainfall hyetograph. The basic assumption behind this method is that water infiltrates into relatively dry soil as a sharp wetting front. The equation of infiltration rate based on this theoretical concept can be expressed as

$$f(t) = K_s + K_s \frac{\left|\psi_f\right|(\theta_s - \theta_i)}{F}, \text{ for } t > t_p \tag{11}$$

$$f(t) = P(\text{b}), \text{ for } t \le t_p \tag{12}$$

where $K_s$ is the saturated hydraulic conductivity (cm/h); $\psi_f$ is the suction head at the wetting front (cm); $\theta_s$ is the saturated moisture content; $\theta_i$ is the initial moisture content before infiltration began; $F$ is the cumulative amount of infiltrated water (cm); $P$ is the rainfall rate (cm/h). With the assumptions and physical structure of this approach, soil characteristics are taken into account. In the simulation cases applied to the study watershed, $K_s$ was set as 1.09 cm/h; $\psi_f$ was set as 11 cm and $\theta_s$ was set as 0.45 according to the soil type of sandy loam.

## 3. Analysis of Time-Interval Restrictions

To examine the magnitude of the time step subject to these two restrictions in various terrains, the Komarovsky watershed, a subwatershed of the Komarovka River Basin in the Far East of Russia, was adopted as the study site. The topography and geographic location of the watershed are shown in Figure 3: it consists of steep hillsides, a higher order of stream networks, and wide riparian areas. The drainage area is 60.3 km$^2$ with an average slope of 0.13. This watershed is mainly covered by forest and only part of the area is alluvial plain in valleys. The soil type is a mixture of sand, loam, and clay. Brown forest soil 25 cm in thickness is predominant in this region [24]. Figure 4 shows the spatial distribution of the local slope in the watershed, in which there is a wide range between 0.001 and 0.8. As shown in Figure 3, there are two rain gauges installed in the Komarovsky watershed, and a flow gauging station at the watershed outlet provides hourly flow data. The annual rainfall in the Komarovsky watershed ranges from 600 mm to 920 mm and the annual evapotranspiration varies with terrain between 450 mm and 550 mm. Because the Komarovsky is an upstream subwatershed of the Komarovka River Basin, detailed channel cross-sectional data is unavailable. Hence, DEM-based overland-flow routing should be among the promising means for runoff simulation. In performing the simulations, a DEM dataset of 10 m resolution was used and the D8 flow direction method was applied.

Considering the land surface of the Komarovsky watershed is mainly forested, the Manning's roughness assigned for the overland flow routing was preliminary inferred in a range of 0.4–0.8 according to the reference value provided by the Hydrologic Engineering Center [25]. In this study, the root-mean-square error (RMSE) between flow records and simulated results was applied to derive an adequate roughness parameter by using an iterative method to evaluate 12 rainfall events. After a series of calibration steps using hydrological records, the overland roughness was selected as 0.62 for the runoff simulations. Figure 5 shows the simulated discharge hydrographs compared to the flow records of four storm events that occurred in the watershed. Basically, the numerical model using the conventional explicit scheme (first-order backward finite-difference method) is capable of providing a fair discharge estimation. Hence, the applicability of the non-inertia wave model can be recognized. In the following sections, the temporal and spatial distributions of the two time-interval restrictions are analyzed and compared in detail.

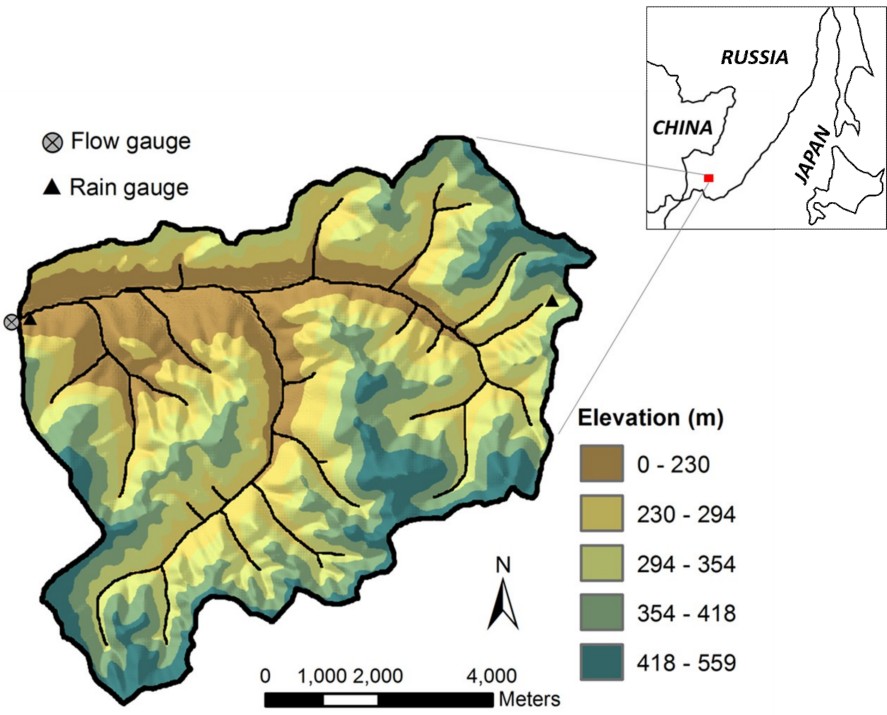

**Figure 3.** Topography and geographic location of the Komarovsky watershed in the far eastern Russia.

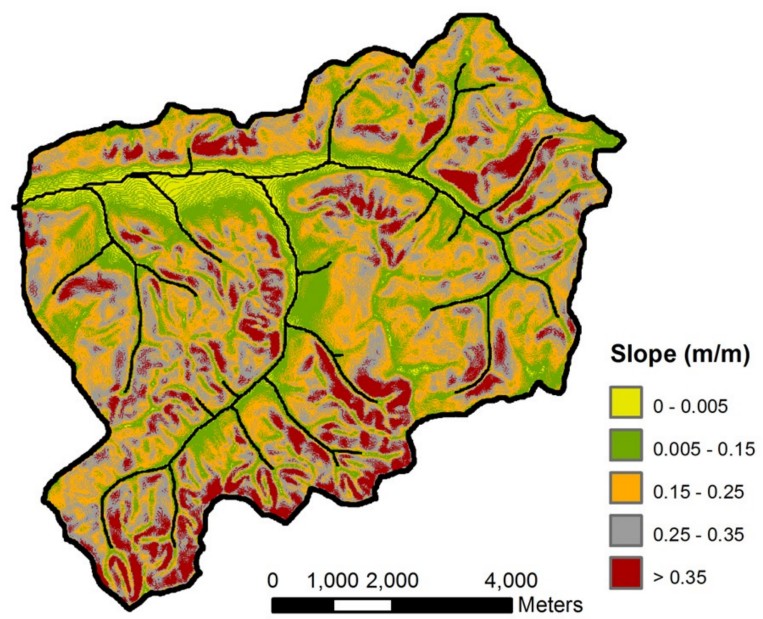

**Figure 4.** Distribution of ground slopes in the Komarovsky watershed.

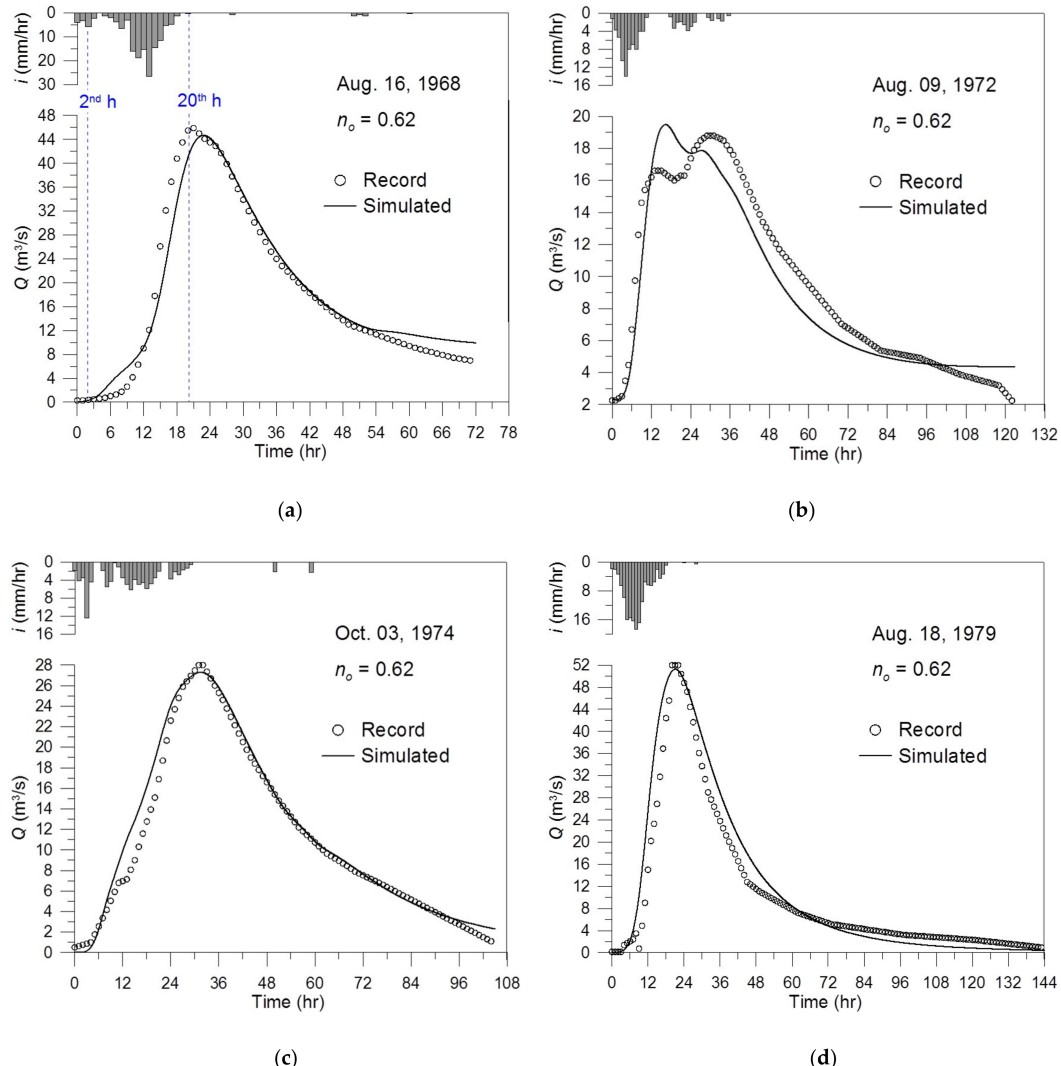

**Figure 5.** Simulated flow hydrographs of four storm events that occurred in the Komarovsky watershed, (**a**) 16 August 1968, (**b**) 9 August 1972, (**c**) 3 October 1974, (**d**) 18 August 1979.

### 3.1. Variation in Time Step During Different Periods of the Storm

As shown in Figure 5a, the storm event that occurred in 1968 was taken to analyze the temporal variation in the two time-interval restrictions during simulation. Water depths and time-interval restrictions occurring at two specific times, the 2nd hour and 20th hour, were investigated. The 2nd hour is the time at the beginning of the storm and the 20th hour is the recorded peak time of the flood. The distribution of the water depth shown in Figure 6a indicates that at the beginning of the storm (during the 2nd hour), the regions with greater water depths ($h \geq 0.08$ m) were scattered on the upstream lower-order streams and the downstream riparian areas. As shown in Figure 6b, with the stream network collecting more rainwater provided by the main storm, the water depth in the mainstream during the 20th hour ($h \geq 0.3$ m) was obviously greater than that presented during the 2nd hour.

In the numerical simulation of the storm that occurred on 16 August 1968, the most limited time step found in the watershed subject to the CFL condition and the Hunter condition was adopted as the computing time interval at each routing round. To investigate the regions prone to cause an overall limitation of computational efficiency, Figures 7 and 8 show the spatial distributions of the allowable time step at the 2nd and 20th hour, respectively, subject to the two conditions. Regarding the CFL condition, as shown in Equation (9), the time step depends on the flow wave speed. Because

at the beginning of the storm, the wave speed is significantly more influenced by the steep terrain than by the depth of the sheet flow, more grids with a time step shorter than 30 s at the 2nd hour are mainly distributed on the low-order steep streams with rapid flow (as shown in Figure 7a. In contrast, at the 20th hour the grids with a time step shorter than 10 s are found in the mainstream as shown in Figure 7b. Such a spatial distribution of time steps subject to the CFL condition is caused by the obvious increase in the wave speed in the mainstream.

Regarding the Hunter condition, which mainly depends on the water depth and the gradient of the water surface level as shown in Equation (10), the distributions of the allowable time step at the 2nd and 20th hours are shown in Figure 8. It can be seen that the change in time-interval restriction between the two specific times is basically similar to that of the CFL condition shown in Figure 7. However, at the 2nd hour only a minority of grids have time steps shorter than 30 s according to the Hunter condition. It can be inferred that at the beginning of the storm, the CFL condition would have a more stringent constraint on time step than that of the Hunter condition. Relatively, at the 20th hour, obviously increased grids with time steps shorter than 1 second appear in the mainstream as shown in Figure 8b. Hence, it can be summarized that the time step is mainly restricted by the CFL criterion at the beginning of the storm, and then the Hunter criterion would alternatively dominate the constraint on time step with the development of water depths in the stream network after receiving more rainwater during the period of the storm.

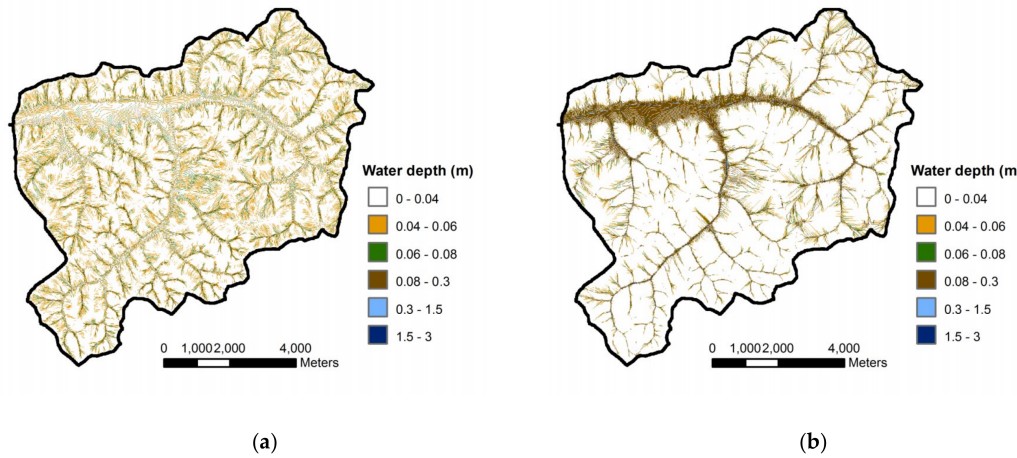

(**a**)                                                                                  (**b**)

**Figure 6.** Distribution of the water depths during the simulation of the storm occurring on 16 August 1968, (**a**) 2nd hour, (**b**) 20th hour.

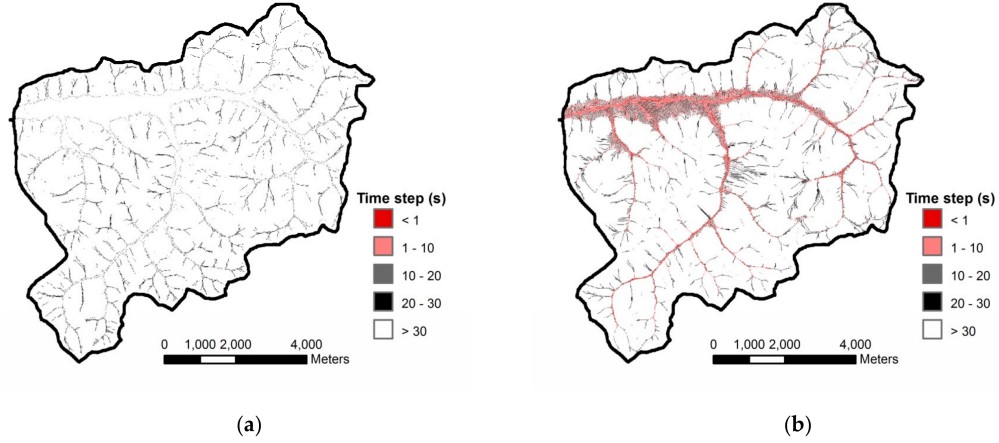

(**a**)                                                                                  (**b**)

**Figure 7.** Distribution of the allowable time steps subject to the CFL condition during the simulation of the storm occurring on 16 August 1968, (**a**) 2nd hour, (**b**) 20th hour.

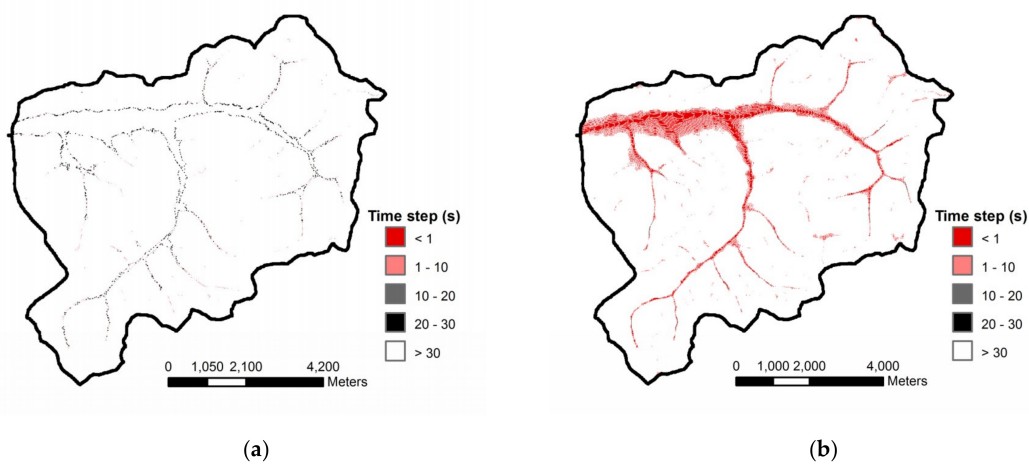

(**a**)　　　　　　　　　　　　　　　　　　　　(**b**)

**Figure 8.** Distribution of the allowable time steps subject to the Hunter flow limit condition during the simulation of the storm occurring on 16 August 1968, (**a**) 2nd hour, (**b**) 20th hour.

*3.2. Variation in Time Step Under Different Topographic Conditions*

In addition to investigating the spatial distribution of the allowable time step during different storm periods, a variation in the two time-step restrictions under different topographic conditions was also analyzed. Three specific zones in the Komarovsky watershed were selected for investigation according to local topographic characteristics. As shown in Figure 9, Zone A is a small steep hillslope (A = 0.61 km$^2$) in which the average slope reaches 0.37 and no distinct channel or gully is apparent. The outlet of Zone A is at a source point of a 1st-order stream. Relatively, Zone B possesses a larger contributing area of approximately 13.58 km$^2$, containing hillsides as well as an obvious 2nd order stream network; and the outlet of Zone B is that of the 2nd-order stream. Zone C is approximately 5.20 km$^2$ in area, denoting downstream riparian areas with a fairly mild slope of less than 0.005; therefore, the extent of Zone C was determined by choosing the grids whose slopes were smaller than 0.005 in the downstream lowland.

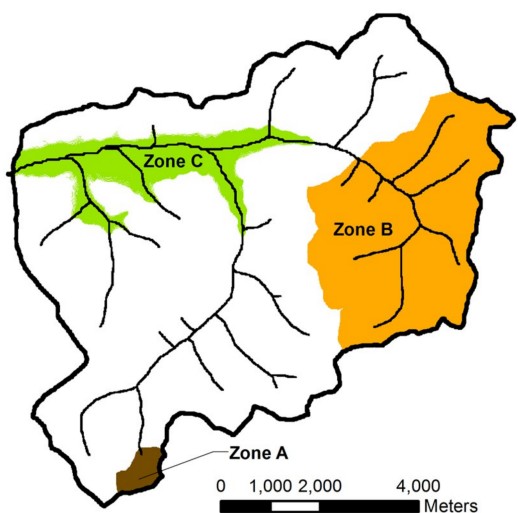

**Figure 9.** Three types of areas with various topographic conditions adopted for the time step analysis.

By applying the rainfall event that occurred on 16 August 1968 (as shown in Figure 5a, the variations in the allowable time steps, subject to the aforementioned restrictions, in each zone during the storm were calculated as shown in Figure 10. As shown in Figure 10a, the steep terrain of Zone A enforces a time step (subject to the CFL condition) quickly decreasing to shorter than 20 s. In contrast, during the first six hours of the simulation, the allowable time step extracted in the riparian area of

Zone C maintains a significantly greater value of approximately 77 s as a result of its fairly mild surface, where the time step (subject to the CFL condition) is less restricted. Moreover, the variation in the time step in Zone A is not as great as that found in the other two zones during the whole storm. The reason for this situation is that the drainage area of Zone A is quite small (only 0.61 km$^2$); therefore, a greater water depth cannot form on the hillslope to cause a continuous reduction in the time step according to the CFL criterion. However, owing to the presence of mainstream in Zone B and Zone C, which are capable of collecting floodwater from upstream areas, the magnitudes of the time step subjected to the CFL condition continuously decreased to 1.1 s and 0.72 s, respectively. Moreover, it was found that the value of the time step would rebound during the period of runoff recession.

Figure 10b shows the results of a similar test considering the Hunter criterion. The time step subject to this criterion in Zone A is longer than 65 s during the entire period of simulation because of the steep terrain, which would cause the larger gradient in the water surface level. Relatively, the allowable time steps in Zone B and Zone C gradually decrease to 0.11 s and 0.03 s during the flood rising period. It should be noted that the limitation of the time step subject to the Hunter condition in Zone B and Zone C is more stringent than that subject to the CFL condition (as shown in Figure 10a).

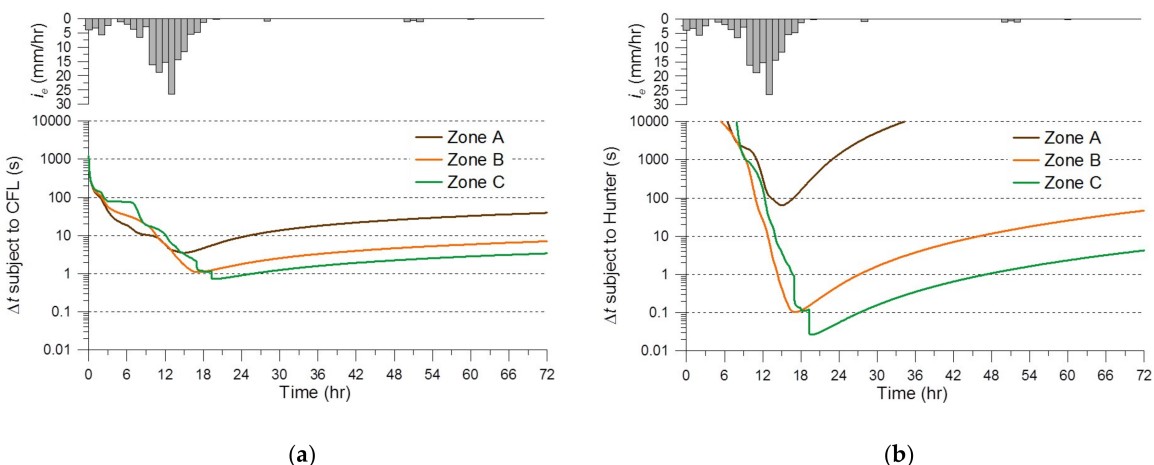

**Figure 10.** Temporal variations of the allowable time steps within different zones during the simulation of the storm occurring on 16 August 1968, (**a**) CFL criterion; (**b**) CFL criterion.

### 3.3. Variation in the Time Step along the Flow Path

Another test was conducted to track the variation in the time step subject to the two criteria along the longest flow path from the most remote point of the watershed to the outlet. As shown in Figure 11, the flow path (as shown by the light blue line) starts from the most remote overland area and connects to the 1st-order, 2nd-order, 3rd-order, and 4th-order streams and the outlet with a total length of 14,073 m. Figure 12 shows the longitudinal elevation and the greatest water depth at each grid along the flow path during the storm that occurred on 16 August 1968 (as shown in Figure 5a). Figure 12a shows that the elevation of the ground bed quickly decreases from 558 m to 345 m within a distance of 1200 m, in which the average slope is approximately 0.18. This terrain condition causes the water depth to significantly change and reach more than 0.5 m in a short distance as shown in Figure 12b. Because the key feature of the stream is its superiority in collecting flow, the water depths in the mainstream area are obviously greater than that in upstream overland areas.

Figure 13 shows the variations in time step, respectively subject to the CFL condition and Hunter condition along this drainage course. For both criteria, the magnitude of the allowable time step is very sensitive to the slope of the ground bed; therefore, the oscillation of the time step along the stream can be clearly determined. Moreover, the variation range in the time step under the Hunter criterion (from $10^{-3}$ s to $10^7$ s) is greater than that under the CFL criterion (from $10^{-1}$ s to $10^2$ s) along the flow path. This figure also shows that the CFL criterion dominates the restriction of the time step in the

upstream 1st-order stream because of the steep topography with rapid and shallow flow; the allowable time step subject to the Hunter criterion gradually reduces to shorter than 1.0 s when the flow path reaches the downstream higher-order streams. Hence, the Hunter criterion would alternately be responsible for the decrease in computational efficiency.

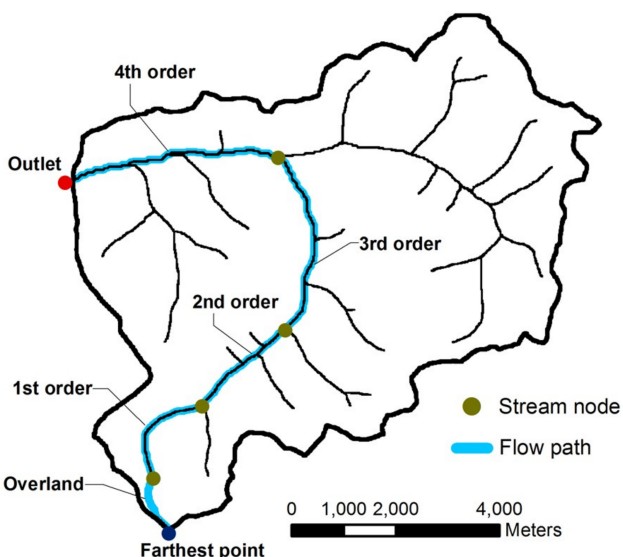

**Figure 11.** The longest drainage path from the farthest watershed ridge to the outlet.

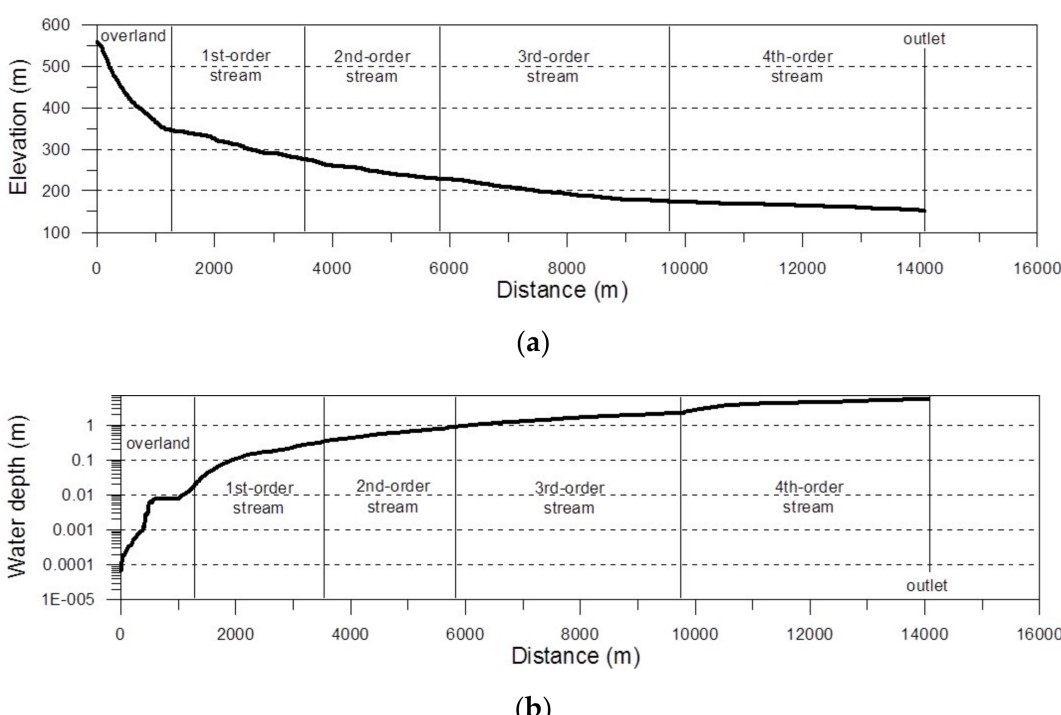

**Figure 12.** Ground bed elevation and the greatest water depths along the longest drainage path during the simulation of the storm occurring on 16 August 1968, (**a**) Elevation of ground bed; (**b**) Greatest water depth.

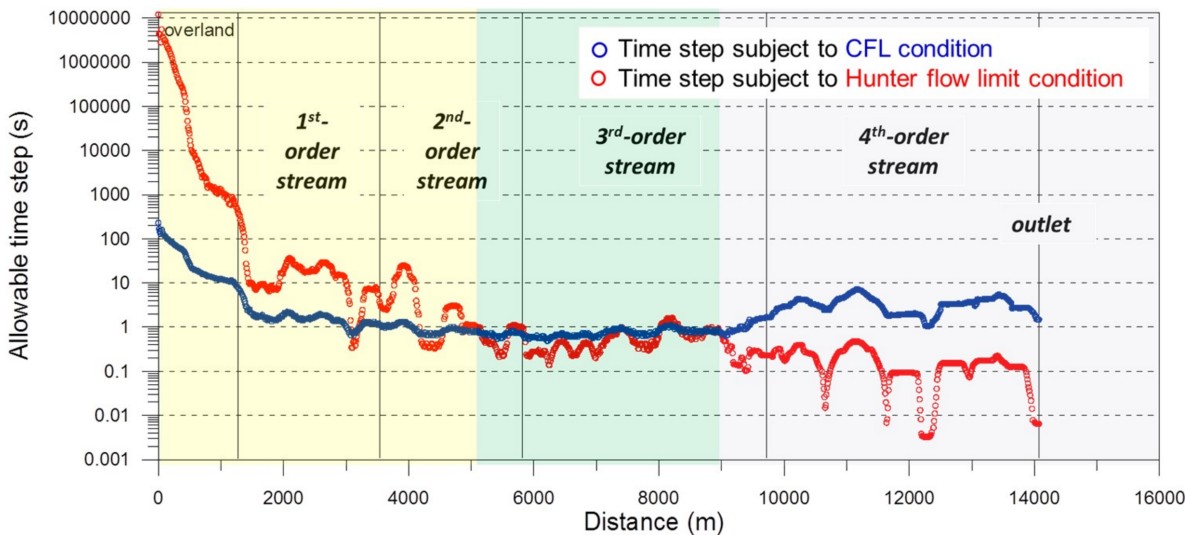

**Figure 13.** Allowable time steps subject to the two different criteria along the longest drainage path during the simulation of the storm occurring on 16 August 1968.

## 4. Review for Modified Algorithms

According to the analyzed results discussed in the previous sections, it can be inferred that only if the two time-step criteria are simultaneously relaxed, can the computational efficiency be effectively enhanced. One may wonder how much computational time can be saved if the two restrictions are relaxed in the simulations. In the following sections, the two modified approaches, separately used to relax the CFL and the Hunter conditions, are first introduced. Then, the execution procedure of each algorithm and how to integrate the two modified methods into a conventional algorithm are explained.

### 4.1. Quasi-2D MacCormack Recursive Formulation

To enlarge the computational time step without being restrained by the CFL criterion, MacCormack [15] preliminarily developed a 1D recursive formulation to ameliorate the numerical stability as follows:

$$\Delta h'_j = \frac{\Delta h_j + \lambda \frac{\Delta t}{\Delta x} \Delta h'_{j,IN}}{1 + \lambda \frac{\Delta t}{\Delta x}} \tag{13}$$

where $\Delta h_j$ is obtained from the continuity equation shown in Equation (5); $\Delta h'_j$ represents the modified form of $\Delta h_j$; $\Delta h'_{j,IN}$ denotes the modified time-varying increment of inflow depth at grid $j$; and $\lambda$ is a parameter that can be defined as follows:

$$16 \text{ August } 1968, \ \lambda = a \cdot Max\left(0, \ c - \frac{\Delta x}{\Delta t}\right) \tag{14}$$

where $c$ is the wave celerity and $a$ is an adjustable coefficient. Huang and Lee [17] showed that the coefficient $a$ mainly depends on the wave celerity and geomorphologic features of the watershed; they found that the most adequate value of $a$ is approximately 0.6 for a non-inertial wave model. Moreover, it should be noted that the variable $\Delta h'_{j,IN}$ shown in Equation (13) cannot be calculated because every single grid in a DEM dataset can be connected to multiple upstream grids. Given this, Huang and Lee [17] converted $\Delta h'_{j,IN}$ into a function of total inflow discharge via Manning's equation as follows:

$$\Delta h'_{j,IN} = h'_{j,IN}(t + \Delta t) - h'_{j,IN}(t) \tag{15}$$

$$= \left( \frac{n}{\sqrt{S_f}} q_{j,IN}(t + \Delta t) \right)^{3/5} - \left( \frac{n}{\sqrt{S_f}} q_{j,IN}(t) \right)^{3/5} \tag{16}$$

As shown in Equation (13), the total inflow discharge $q_{j,IN}(t + \Delta t)$ is needed to perform the recursive formulation shown in Equation (11), which is conducted to permit a larger time step. To obtain the total inflow discharge from upstream grids prior to performing Equation (13), Huang and Lee [17] suggested a valid routing sequence, following the descending order of a water stage at each grid for the DEM-based runoff simulation. The recursive formulation shown in Equation (13) is implemented to derive $\Delta h'_j$ by correcting $\Delta h_j$, which is calculated from the continuity equation. Subsequently, the water depth at $t + \Delta t$ can be calculated as follows:

$$h_j(t + \Delta t) = h_j(t) + \Delta h'_j \tag{17}$$

### 4.2. Bates Inertial Momentum Formulation

Hunter et al. [16] proposed an approach to avoid flow reflux between adjacent grids by adding the local acceleration term into the non-inertia wave equation as follows:

$$\frac{\partial q}{\partial t} - gh \left( S_o - \frac{\partial h}{\partial x} \right) + ghS_f = 0 \tag{18}$$

It can be seen that in the aforementioned equation only the convective acceleration term is omitted from the full-type momentum equation. Hunter et al. [16] denoted that the inertial force, the first term (local acceleration term) of Equation (17), considered in the momentum equation can alleviate excessive flux delivery to the downstream grid. Based on this research finding, Bates et al. [8] further exploited Equation (17) to deduce an inertial momentum formulation as follows:

$$q_j(t + \Delta t) = \frac{q_j(t) + gh_j(t + \Delta t) \cdot \Delta t \left( S_o - \frac{\partial h_j(t)}{\partial x} \right)}{1 + gh_j(t + \Delta t) \cdot \Delta t \frac{n^2 q_j(t)}{h_j^{10/3}(t + \Delta t)}} \tag{19}$$

Bates et al. [8] reported that this modified momentum equation, used to replace Equation (8) (the non-inertia wave equation), is capable of relaxing the Hunter condition owing to its superior convergence. Nevertheless, the restraint of the time interval subject to the CFL condition remains when applying the aforementioned scheme [8].

### 4.3. Execution Procedures of the Modified Models

In the preceding two sections, the quasi-2D MacCormack recursive formulation was devised to modify the water depth that derived from the continuity equation, and the Bates inertial momentum formulation was adopted to replace the momentum equation that was originally based on the non-inertia wave assumption. The following content illustrates the routing procedure of combining the two aforementioned methods to simultaneously relax the CFL condition and the Hunter condition [18].

1.  Assign the rainfall condition and the initial condition of the ground bed;
2.  Arrange the routing sequence of each grid according to the descending order of the water stage;
3.  Determine the steepest flow direction of each grid;
4.  Derive the time-varying increment of water depth $\Delta h_j$ using Equation (5);
5.  Calculate the related parameters used in the recursive formulation according to Equations (14) and (15);
6.  Modify the increment of the water depth using Equation (13);
7.  Calculate the water depth at following Equation (16);

8. Calculate the discharge by substituting the updated water depth into Equation (18);
9. Accumulate the flow discharge for the downstream grid;
10. Repeat the process from Step (2) to Step (9) in the next time step.

To compare and evaluate the computational performance of different numerical approaches, the routing procedures of various methods are listed in Table 1.

**Table 1.** Comparison of routing procedures using different numerical methods.

| Routing Procedure | Numerical Methods | | | |
| --- | --- | --- | --- | --- |
| | Conventional | Bates | MacCormack | Integrated |
| **Assign the rainfall condition and the initial condition** | √ | √ | √ | √ |
| Sort the routing sequence of each grid | | | √ | √ |
| Determine the steepest flow direction of each grid | √ | √ | √ | √ |
| Derive the time-varying increment of water depth using Equation (5) | √ | √ | √ | √ |
| Calculate the related parameters using Equations (14) and (15) | | | √ | √ |
| Modify the increment of water depth using Equation (13) | | | √ | √ |
| Calculate the water depth using Equation (16) | √ | √ | √ | √ |
| Calculate the flow discharge　using Equation (8) | √ | | √ | |
| 　　　　　　　　　　　　　　using Equation (18) | | √ | | √ |
| Accumulate the flow discharge for the downstream grid | √ | √ | √ | √ |
| CFL condition | Required | Required | No need | No need |
| Hunter condition | Required | No need | Required | No need |

## 5. Benchmark Test of Runoff Simulation

A simulation case for benchmark test is conducted to demonstrate the applicability of three aforementioned modified algorithms, including the MacCormack recursive formulation, Bates inertial momentum formulation and the integrated algorithm by comparing the simulated hydrographs with the analytical solution. In this study, to obtain an analytical solution for model comparison, runoff simulation on an impervious overland plane was implemented to examine the numerical accuracy of each model. Continuous rainfall with a constant intensity of 70 mm/h was assigned on an 800 m overland plane, of which the uniform slope is 0.3 and the size of mesh spacing is 2 m. In performing the numerical tests, a fixed Manning's roughness coefficient of 0.1 was assigned and all the simulations were executed commencing at an initially dry surface.

Figure 14a,b show the simulated hydrographs of discharge and water depth at the end of the plane by applying the four different algorithms based on diffusion-wave approximation and the analytical solution for kinematic-wave equation. Several previous studies have indicated that the flow transport on steep terrains simulated by the kinematic wave equation can be analogous to that simulated by either the diffusion-wave or dynamic-wave equations [26–28]. In the numerical tests, the maximum allowable time step was 2 s when the conventional algorithm and the Bates inertial momentum formulation were performed, while the time step can be enlarged to 30 s when the MacCormack recursive formulation and the integrated algorithm were adopted owing to the efficacy of releasing the CFL condition. It should be noted that the situation of flow refluxing would not occur in the case of a 1D overland plane, hence, the CFL condition was the dominant criterion to restrain the time step. As shown in Figure 14a,b, both the discharge and water depth hydrographs predicted by the four algorithms are close to the analytical solution of the kinematic-wave equation despite the slight delay of time to equilibrium. The hydrographs generated by the integrated algorithm and the Bates inertial momentum formulation are similar because of using the identical momentum equation as shown in Equation (17), in which one more inertial force term is included in the original diffusion wave equation as shown in Equation (4). Hence, the larger deviations between the simulated results and the analytical solution of kinematic-wave equation can be found in these two methods in comparison with the other two numerical solutions of non-inertial wave equation. Basically, this test manifests the feasibility of applying different modified models for rainfall-runoff simulation.

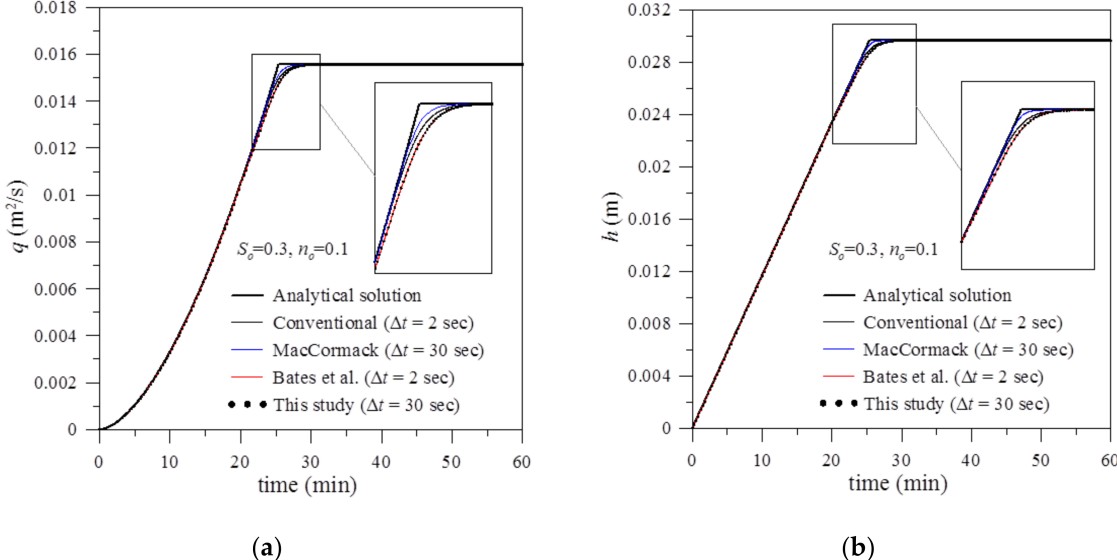

**Figure 14.** Predicted and analytical hydrographs using different algorithms, (**a**) Discharge, (**b**) Water depth.

## 6. Evaluation of Computational Efficiency

To clarify the efficiency of relaxing the time-step restrictions by applying modified approaches in runoff simulations, the hydrological records of four storm evens as shown in Figure 5 were adopted to compare computational costs (CPU time) among the different algorithms. As shown in Table 1, "conventional" denotes the conventional explicit scheme (first-order backward finite difference method), in which the CFL condition and Hunter condition are simultaneously required; hence the most limited time step among all the grids in the watershed should be sought at each routing round and selected as the computational time step. To meet the two restrictions while conducting the conventional explicit scheme, the time step $\Delta t$ was found to vary in a range of 0.001–20 s. "MacCormack" denotes the algorithm that combines the quasi-2D MacCormack recursive formulation with the conventional explicit scheme to relax the CFL condition; hence, only the Hunter criterion is required. "Bates" denotes the method that replaces the non-inertia wave equation Equation (8) with the Bates inertial momentum formulation Equation (16) in the conventional explicit scheme to relax the Hunter condition, but the CFL criterion is still required. "Integrated" denotes the algorithm that integrates both the quasi-2D MacCormack recursive formulation and the Bates inertial momentum formulation into the conventional explicit scheme; hence, the computational time step $\Delta t$ can be lengthened to 60 s without being subject to the CFL condition and Hunter condition.

Although in this case study the integrated algorithm still has its maximum limit on time step (60 s) when applied to the Komarovsky watershed, Table 2 shows that the integrated algorithm obviously needs less CPU time than that of the other three algorithms for all of the four storm events. It also shows that individually applying the quasi-2D MacCormack recursive formulation or the Bates inertial momentum formulation in the model is not sufficient to enhance the overall performance because one of the time-interval restrictions remains. To clarify the computational efficiency using different numerical algorithms, Table 3 shows the ratios of CPU time, obtained by comparing each of the earlier methods to the integrated algorithm. The results show that the computational speed of the integrated algorithm outperforms that of the conventional explicit scheme by a maximum factor of 67 among the four storm events. Moreover, the integrated algorithm is more than 39 and 17 times as fast as the two methods, individually applying the quasi-2D MacCormack recursive formulation and the Bates inertial momentum formulation. However, as shown in Table 4, an index, normalized mean square error (NMSE), was adopted to calculate the deviations in the discharge hydrographs between the conventional and modified methods. Results show that the integrated model produces larger

deviations because the calculated time step is lengthened to 60 s. However, the NMSE values in the four simulation sets, assigned with different rainfall conditions, are still within $10^{-2}$, and do not present an obviously visible deviation among the hydrographs.

**Table 2.** Comparison of computational time among different algorithms.

| Date of Storm Event | CPU Time (min) | | | |
|---|---|---|---|---|
| | Conventional ($\Delta t = 0.001-20$ s) | MacCormack ($\Delta t = 0.001-20$ s) | Bates ($\Delta t = 0.1-20$ s) | Integrated ($\Delta t = 60$ s) |
| 16 August 1968 | 1778.81 | 1225.89 | 568.86 | 28.83 |
| 9 August 1972 | 2543.27 | 1926.28 | 836.75 | 48.85 |
| 3 October 1974 | 2258.56 | 1715.66 | 742.18 | 41.64 |
| 18 August 1979 | 3864.42 | 2585.22 | 1246.17 | 57.66 |

**Table 3.** Evaluation of computational efficiency for different algorithms.

| Date of Storm Event | Ratio of CPU Time | | |
|---|---|---|---|
| | Conventional/Integrated | MacCormack/Integrated | Bates/Integrated |
| 16 August 1968 | 61.7 | 42.5 | 19.7 |
| 9 August 1972 | 52.1 | 39.4 | 17.1 |
| 3 October 1974 | 54.2 | 41.2 | 17.8 |
| 18 August 1979 | 67.0 | 44.8 | 21.6 |

**Table 4.** Deviation of hydrographs between the conventional and modified algorithms.

| Date of Storm Event | NMSE (Normalized Mean Square Error) | | |
|---|---|---|---|
| | MacCormack vs. Conventional | Bates vs. Conventional | Integrated vs. Conventional |
| 16 August 1968 | $2.47 \times 10^{-3}$ | $3.34 \times 10^{-3}$ | $4.82 \times 10^{-3}$ |
| 9 August 1972 | $1.83 \times 10^{-3}$ | $2.16 \times 10^{-3}$ | $5.25 \times 10^{-3}$ |
| 3 October 1974 | $4.59 \times 10^{-3}$ | $6.85 \times 10^{-3}$ | $8.77 \times 10^{-3}$ |
| 18 August 1979 | $4.73 \times 10^{-3}$ | $5.91 \times 10^{-3}$ | $6.42 \times 10^{-3}$ |

Note: NMSE $= \frac{1}{T} \sum\limits_{t=1}^{T} \frac{(Q_t - Q_{conventional})^2}{\overline{Q}\,\overline{Q}_{conventional}}$, $\overline{Q} = \frac{1}{T} \sum\limits_{t=1}^{T} Q_t$ and $\overline{Q}_{conventional} = \frac{1}{T} \sum\limits_{t=1}^{T} Q_{conventional\ t}$.

## 7. Conclusions

As reported in previous studies, there are two types of time-interval restriction regulated to maintain the accuracy and stability during grid-based flood simulations. The CFL criterion is enforced to validate explicit schemes, and the Hunter condition is conducted to alleviate the situation of flow reflux between adjacent grids. This study aimed to detect the spatial and temporal variations in allowable time steps subject to these two criteria under different terrain conditions and runoff states, hence, the inefficient problem in conducting distributed flood simulation can be understood.

Three findings can be summarized via a case study of the Komarovsky watershed in the far East of Russia. First, at the beginning of the runoff simulation, the CFL criterion in the steep hillsides dominates the computational time step. With increasing water depth in the mainstream, the Hunter condition gradually becomes the main reason for the reduction in the time interval. Second, for a steep hillside with a small drainage area, the allowable time step subject to the CFL condition was much shorter than that subject to the Hunter condition during the entire period of simulation; however, for a watershed that contains wide channels and mild riparian areas, the magnitude of the time step subject to the Hunter condition can be below $10^{-1}$ s, which is more restrained than the allowable time step subject to the CFL condition. Third, according to the result of the allowable time steps extracted along the longest drainage path from the watershed ridge to the outlet, the variation extent of the time

step under the Hunter criterion (from $10^{-3}$ s to $10^7$ s) was greater than that under the CFL criterion (from $10^{-1}$ s to $10^2$ s). The CFL criterion provided a more limited time step at the upper reach ahead of the first-order stream node; relatively, for the remainder of the downstream path, the Hunter condition resulted in a shorter time step.

Through an in-depth investigation of time-step limitations, it was shown that only if the two time-interval criteria are simultaneously relaxed can the computational performance be effectively enhanced. This study also emphasized the significance of integrating the two modified approaches (the quasi-2D MacCormack recursive formulation and the Bates inertial momentum formulation) into the conventional explicit algorithm to relax the CFL and Hunter conditions. The integrated algorithm was capable of surpassing the two methods, i.e., individually applying the quasi-2D MacCormack recursive formulation and the Bates inertial momentum formulation, by at least 39 and 17 times in terms of computational efficiency. Consequently, it can be applied to a real-time flood forecasting system to strengthen the numerical stability as well as the calculation speed in runoff simulation.

**Author Contributions:** Conceptualization, P.-C.H. and K.T.L.; methodology, P.-C.H.; software, P.-C.H.; validation, P.-C.H.; K.T.L. and B.I.G.; formal analysis, P.-C.H.; investigation, P.-C.H.; resources, K.T.L. and B.I.G.; data curation, B.I.G..; writing—original draft preparation, P.-C.H.; writing—review and editing, P.-C.H.; visualization, P.-C.H.; and K.T.L.; supervision, K.T.L.; project administration, K.T.L.; funding acquisition, K.T.L. and B.I.G.

**Funding:** This research was funded by Ministry of Science and Technology, Taiwan, grant number 107-2625-M-019-003; Russian Science Foundation, grant number 17-77-30006; Russian Foundation for Basic Research, grant number 19-05-00353.

**Acknowledgments:** Financial supports provided by the Ministry of Science and Technology (MOST), Taiwan, Russian Science Foundation and Russian Foundation for Basic Research are sincerely acknowledged.

**Conflicts of Interest:** The authors declare no conflict of interest.

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
