# Peer review of "Influence of Topographic Characteristics on the Adaptive Time Interval for Diffusion Wave Simulation"

_water, doi:10.3390/w11030431_

Round 1
Reviewer 1 Report
The authors show in this paper a very interesting analysis of the computational efficiency of runoff numerical simulations. This type of simulations are usually carried out starting from Saint Venant’s equations and, therefore, two time-interval restrictions are involved: the Courant-Friedich-Lewy condition (CFL) and the Hunter condition. The former is associated with areas with a high averaged slope and the latter is associated with areas with low averaged slope. In these low slope areas, and under some conditions, refluxing between elements of the numerical grid can be observed. To avoid this effect Hunter condition must be taken into account.
The paper is very interesting, but some issues are not clear to me to accept this work. These issues are:
the authors calibrate the model obtaining a value of roughness equal to 0.62. But this is a “visual calibration”. In my opinion the use a statistic parameter (Nas-Shutclife, PBIAS, RMR....) is necessary to calibrate the model in a correct way;
no details are specified related with the infiltration model: what are the values of the parameters of the Green-Amp model defined by the authors? and why?
a general map of the location of the watershed would be very useful.
But, from my point of view, the main problem of the paper is this: all the figures are not well defined. Some of them appear cut out and none of the captions appear in the paper. This make very difficult to read and understand the paper.
Author Response
RESPONSE TO REVIEWER’S COMMENTS
The authors sincerely thank the associate editor and the anonymous referee for their valuable comments. As to the Reviewer’s suggestions, necessary modifications have been made by the authors. All the amended paragraphs and sentences are marked red in the revised manuscript.
Reply to Reviewer 1
1. The authors calibrate the model obtaining a value of roughness equal to 0.62. But this is a “visual calibration”. In my opinion the use a statistic parameter (Nas-Shutclife, PBIAS, RMR....) is necessary to calibrate the model in a correct way.
Response:
The authors understand the concern about calibrating the model parameter mentioned by the reviewer. In this study, the root-mean-square error (RMSE) between flow records and simulated results was applied to derive an adequate roughness by using an iterative method to evaluate 12 rainfall events. The details have been added in the revised manuscript (page 6, line 178-181). Since the objective of this study is to elaborate the differences between the two time-interval restrictions and to demonstrate the significance of adopting an integrated modified algorithm in order to enhance the computational efficiency, the authors suppose that a conventional manner applied for roughness calibration could be accepted as long as the simulation errors can be within an acceptable range. Absolutely, the selection of a better way for model calibration can be another issue to improve the simulation accuracy.
2. No details are specified related with the infiltration model: what are the values of the parameters of the Green-Ampt model defined by the authors? and why?
Response:
As suggested by the reviewer, descriptions for the soil type and related parameters have been added in the revised manuscript (page 6, line 171-175). With the assumptions and physical structure of the Green-Ampt method, soil characteristics are taken into account. In this study, the saturated hydraulic conductivity was set as 2.60 cm/hr and saturated moisture content was set as 0.45 according to the soil type of sandy loam in the Komarovsky watershed.
3. A general map of the location of the watershed would be very useful.
Response:
The authors thank the reviewer’s suggestion. The indication for the geographic location of the study watershed has been included in Figure 3 of the revised manuscript.
4. From my point of view, the main problem of the paper is this: all the figures are not well defined. Some of them appear cut out and none of the captions appear in the paper. This make very difficult to read and understand the paper.
Response:
The authors sincerely apologize for editing errors of figures and captions. In the revised version, each figure with its caption has been checked and amended accordingly.
Reviewer 2 Report
water-420346: Influence of topographic characteristics on the adaptive time interval for runoff simulation
This paper presents an in-depth investigation of time-step limitations when using diffusion wave model for runoff simulation. The title of this manuscript is very interesting and got me greatly interested. It seems like to discuss on time-step limitations of general hydrodynamic-based runoff simulation models, and this issue is a topic of interest. However, from the content of the paper, the authors used a diffusion wave model for runoff simulation, and only discussed the time-step limitations for diffusion wave model. So both the title and body content would be not sufficiently rigorous.
After all, this manuscript would be useful for readers who adopting diffusion wave models. So I am glade to recommend the paper for publication in the MDPI Water Journal. Some major points that need to be revised are listed as follows.
1. Title. The title should be restricted to diffusion wave model.
2. Abstract and body content. Some sentence should be modified for rigorous presentation. For example, Lines 13-15, the presented two types of time-interval restrictions are only necessarily enforced when using diffusion wave model. As far as I know only CFL condition is used for explicit schemes of hydrodynamic models using fully shallow water equations.
3. Some benchmark test case of runoff simulation should be used for model validation. Please refer ‘V-shape idealized catchment’ adopted by Liang et al. Liang Q, Xia X, and Hou J. Catchment-scale high-resolution flash flood simulation using the GPU-based technology. Procedia Engineering 154 ( 2016 ) 975-981.
4. The titles of figures are all missing. Besides, the pictures are not very sharp and should be redrew.
Author Response
RESPONSE TO REVIEWER’S COMMENTS
The authors sincerely thank the associate editor and the anonymous referee for their valuable comments. As to the Reviewer’s suggestions, necessary modifications have been made by the authors. All the amended paragraphs and sentences are marked red in the revised manuscript.
Reply to Reviewer 2
1. Title. The title should be restricted to diffusion wave model.
Response:
As suggested by the reviewer, the title of this manuscript has been changed to Influence of topographic characteristics on the adaptive time interval for diffusion wave simulation
2. Abstract and body content. Some sentence should be modified for rigorous presentation. For example, Lines 13-15, the presented two types of time-interval restrictions are only necessarily enforced when using diffusion wave model. As far as I know only CFL condition is used for explicit schemes of hydrodynamic models using fully shallow water equations.
Response:
The authors sincerely thank the reviewer’s advice. Sentences related to this issue in the manuscript have been checked and modified accordingly.
3. Some benchmark test case of runoff simulation should be used for model validation. Please refer ‘V-shape idealized catchment’ adopted by Liang et al. Liang Q, Xia X, and Hou J. Catchment-scale high-resolution flash flood simulation using the GPU-based technology. Procedia Engineering 154 ( 2016 ) 975-981.
Response:
As suggested by the reviewer, a section for benchmark test has been included in the revised manuscript (page 14, line 376-401; Figure 14).
4. The titles of figures are all missing. Besides, the pictures are not very sharp and should be redrew.
Response:
The authors sincerely apologize for editing errors of figures and captions. In the revised version, each figure with its caption has been checked and amended accordingly.
Round 2
Reviewer 1 Report
The authors answered properly the suggestions/comments of the reviewer.
Author Response
RESPONSE TO REVIEWER’S COMMENTS
The authors sincerely thank the associate editor and the anonymous referee for their valuable comments. As to the Reviewer’s suggestions, necessary modifications have been made by the authors.
Reviewer 2 Report
The limitation of the quasi-2D model, comparing to the fully 2D model, should be discussed in the Introduction. Does the quasi-2D model can run on unstructured triangular grids?
The Green-Ampt equation and parameters should be presented in the Section 2.
Lines 386 and 397, Figure 5 should be Figure 14.
Figure 14 shows that the proposed model acheive a worst accuracy. Some reasons should be discussed.
The Section 5 of model validation should be put to the front of Section 3.
Author Response
RESPONSE TO REVIEWER’S COMMENTS
The authors sincerely thank the associate editor and the anonymous referee for their valuable comments. As to the Reviewer’s suggestions, necessary modifications have been made by the authors. All the amended paragraphs and sentences are marked red in the revised manuscript.
Reply to Reviewer 2
1. The limitation of the quasi-2D model, comparing to the fully 2D model, should be discussed in the Introduction. Does the quasi-2D model can run on unstructured triangular grids?
Response:
As suggested by the reviewer, the content discussing about the quasi-2D model and its applicability for unstructured grids have been included in the Introduction (page 1, line 41; page 2, line 42-54). As demonstrated in previous studies, the quasi-2D model can be applied on a topographic data with unstructured grids (Kuiry et al., 2010). However, it should be noted that this study only focus on the time-step analyses for diffusion wave model based on structured grids, hence, the applicability of modified algorithm adopted in an unstructured data still needs to be discussed in the future research. Related explanations have also been included in the Introduction (page 2, line 70-73; page 2, line 78-80).
2. The Green-Ampt equation and parameters should be presented in the Section 2.
Response:
As suggested by the reviewer, the theoretical equation and the related parameters for the rainfall infiltration estimation have been moved to section 2.3 (page 5, line 157-170).
3. Lines 386 and 397, Figure 5 should be Figure 14.
Response:
The authors sincerely apologize for the editing error. In the revised version, the figure number has been amended (page 14, line 402; page 15, line 413).
4. Figure 14 shows that the proposed model achieve a worst accuracy. Some reasons should be discussed.
Response:
The authors sincerely thank the reviewer’s suggestion. Explanation for this issue has been included in the revised manuscript (page 15, line 415-421). The hydrographs generated by the integrated algorithm and the Bates inertial momentum formulation are similar because of using the identical momentum equation as shown in Eq. (17), in which one more inertial force term is included in the original diffusion wave equation as shown in Eq. (4). Hence, the larger deviations between the simulated results and the analytical solution of kinematic-wave equation can be found in these two methods in comparison with the other two numerical solutions of non-inertial wave equation.
5. The Section 5 of model validation should be put to the front of Section 3.
Response:
The authors sincerely thank the reviewer’s suggestion, and did think about this arrangement for the manuscript structure. However, since the review of different modified algorithms is discussed in the Section 4 of manuscript, it should be more adequate to put the content of model validation for various algorithms after Section 4. The authors hope the reviewer can be considerate of this difficulty.
